# Fermented Cosmetics and Metabolites of Skin Microbiota—A New Approach to Skin Health

**Akira Otsuka [1], Chihiro Moriguchi [2], Yuka Shigematsu [1], Kurumi Tanabe [1], Nanami Haraguchi [1], Sonoko Iwashita [1], Yoshihiro Tokudome [1,3] and Hiroshi Kitagaki [1,2,*]**

1 Graduate School of Health Sciences, Saga University, Honjo-cho, 1, Saga 840-8502, Japan
2 Faculty of Agriculture, Saga University, Honjo-cho, 1, Saga 840-8502, Japan
3 Laboratory of Cosmetic Sciences, Regional Innovation Center, Saga University, Honjo-cho, 1, Saga 840-8502, Japan
* Correspondence: ktgkhrs@cc.saga-u.ac.jp

**Abstract:** The skin covers our entire body and is said to be the "largest organ of the human body". It has many health-maintaining functions, such as protecting the body from ultraviolet rays and dryness and maintaining body temperature through energy metabolism. However, the number of patients suffering from skin diseases, including atopic dermatitis, is increasing due to strong irritation of the skin caused by detergents that are spread by the development of the chemical industry. The skin is inhabited by about $10^2$–$10^7$ cells/cm$^2$ and 1000 species of commensal bacteria, fungi, viruses, and other microorganisms. In particular, metabolites such as fatty acids and glycerol released by indigenous skin bacteria have been reported to have functional properties for the health of the skin. Therefore, skin-domesticating bacteria and the metabolites derived from those bacteria are used in many skincare product ingredients and function as probiotic cosmetics. Japanese traditional fermented stuff, used as foods in Japan for over 1300 years, are now being applied as fermented cosmetics. Fermented cosmetics are expected to have multifaceted health functionality and continue to grow as products in the natural skincare product market. In this review, we consider approaches to skin health using fermented cosmetics and modulation of skin microflora metabolites.

**Keywords:** fermentation; skin microbiota; cosmetics





## 1. Structure of Skin

The skin covers our entire body and is said to be the "largest organ of the human body". It covers an area of about 1.6 square meters (adult) and comprises approximately 15–16% of the body weight [1,2]. The skin consists of three layers (epidermis, dermis, and subcutaneous tissue) and skin appendages.

The epidermis, the outermost layer of the skin, has an average thickness of 0.2 mm and consists of four layers: basal layer, spinous layer, granular layer, and stratum corneum (five layers, including a transparent layer only on the palms and soles) ([3], pp. 13–20).

The basal layer, the lowest layer of the epidermis, is the only layer of epidermal cells capable of cell division. New cells are created in this layer and are pushed up to the upper layers through repeated cell division. The cells in the stratum corneum are eventually exfoliated and peeled off. This cycle is called turnover and when it is in order, healthy skin is maintained [4]. In addition, the basement membrane, which exists in the lowest part of the skin, serves as a connection with the dermis and exchanges waste products and nutrients. The spinous layer forms the thickest layer (8–10 layers), and comprises the majority of the epidermis. On the lower side, polygonal cells become flattened as they ascend. The stratum granulosum consists of one or two flattened layers of cells; the keratohyalin granules present in this layer prevent ultraviolet light from entering the skin and being absorbed deeply. The stratum corneum, the outermost layer of cells, serves as the boundary between the inside and outside of the organism, preventing bacteria, UV

rays, and other chemicals from entering the body from the outside and retaining water within the body [5,6].

The dermis averages 1 to 2 mm in thickness and consists of three layers: papillary layer, subpapillary layer, and reticular layer. Blood vessels, nerves, and lymphatic vessels pass through it, with mast cells related to inflammation and histiocytes related to immunity also present. A jelly-like matrix called hyaluronic acid is laid down between the fibrous proteins called collagen and elastin. Since blood vessels are not distributed in the epidermis, capillaries in the dermis provide nutrients, oxygen, and water to the epidermis, carry away waste products and carbon dioxide, and support cell division [5], ([6], pp. 27–29).

The thickness of the subcutaneous tissue averages from 4 to 9 mm. It is mostly composed of adipose tissue, also known as subcutaneous adipose tissue. Connecting the skin to muscles and bones, it acts as a buffer to protect them from external forces. It also functions to metabolize energy and maintain body temperature ([6], pp. 29–32).

Skin appendages include hair, hair follicles, sweat glands, and sebaceous glands [5], ([6], pp. 29–32).

## 2. Skin Diseases in the Modern Era

The frequency and severity of skin diseases vary, depending on the influence of the living environment and differences in skin physiology of sweating and sebum secretion due to age and gender.

### 2.1. Eczema Group (Atopic Dermatitis and Contact Dermatitis)

According to a survey by the Japanese Dermatological Association [7], the "eczema group" accounts for the highest percentage of dermatological consultations (38.85%) and is more common in all age groups.

There is little difference between male and female patients with atopic dermatitis, while contact dermatitis is more common in females. When limited to the face and neck area, women are about six times more likely than men to have contact dermatitis. About one third of cases are probably caused by cosmetics [8]. Atopic dermatitis is a "multifactorial" disease caused by a combination of constitutional factors, such as atopic predisposition (predisposition to allergies), skin conditions with impaired barrier function, and environmental factors including allergens and external stimuli to the skin. Symptom trends and sites of onset change with age. Contact dermatitis is caused by substances such as cosmetics, pharmaceutical ingredients, metal ornaments, clothing, animals, and plants that are frequently touched in daily life.

### 2.2. Skin Cancer (Malignant Tumor)

Relatively common among the elderly, skin cancer generally develops on areas exposed to sunlight [9]. Since any cell in the epidermis can become cancerous, there are many types of skin cancer. The most common are basal cell carcinoma, squamous cell carcinoma, and malignant melanoma (melanoma), and the precancerous conditions are Bowen's disease and Paget's disease. Ultraviolet light, trauma, radiation, and viruses are thought to be the main causes of these diseases, although this is unclear.

### 2.3. Acne (Acne Vulgaris)

When the epidermis does not turn over normally, the keratin in the pores thickens, the pore outlet becomes blocked, and sebum clogs the pores, causing overgrowth of acne bacteria that feed on sebum, resulting in inflammation. It is said to be caused by an increase in androgen, a male hormone that increases sebum secretion, during puberty [10,11].

Our skin is exposed to an even harsher environment due to the spread of detergents with high cleaning power, resulting from the development of the modern chemical industry and the drying of our living environment. In addition, due to the spread of coronavirus infection, antibacterial and sterilizing goods are widely used around the world, and an

increasing number of people are washing their skin excessively. Excessive cleaning not only removes dirt but also necessary sebum and indigenous skin bacteria.

In the modern day, people should understand the structure of the skin and the causes of skin diseases, perform skincare appropriate to one's gender, age, physical activity status, and body parts, and keep the skin clean for good health.

## 3. Types of Indigenous Skin Bacteria and Their Effects on Health

### 3.1. Indigenous Skin Bacteria

The human skin harbors approximately $10^2$–$10^7$ cells/cm$^2$ per person and 1000 species of bacteria, fungi, viruses, and other microorganisms [2,12]. The skin is in direct contact with the outside world. Therefore, aerobic bacteria are mainly found on the surface layer, while anaerobic bacteria inhabit the hair follicles and sebaceous glands as commensal bacteria. *S. epidermidis*, *S. aureus*, and *Micrococcus*, which are aerobic, and *Cutibacterium acnes* (*anaerobic bacilli*) are the most widespread [13].

Although less abundant, the fungi *Malassezia*, *Candida*, and *Trichophyton* are also present.

Representative indigenous skin bacteria include *S. epidermidis*, *Micrococcus* genus bacteria, *C. acnes*, and a type of yeast [14].

### 3.2. Distribution of Indigenous Skin Bacteria

The composition of the microbial layer of human skin varies by site, as UV exposure, pH, temperature, moisture, sebum content, and topography differ depending on the location. The types of microorganisms indigenous to the skin were recently investigated by the U.S. National Human Genome Research Institute [15]. The results showed that the microorganisms with the highest abundance were bacteria. In seborrheic areas (forehead, back, etc.), bacteria of the genus *Cutibacterium*, such as Acne bacillus, were found, while in dry areas (forearms and buttocks), bacteria of the phylum *Actinobacteria*, *Proteobacteria*, *Firmicutes*, and *Bacteroidetes* were most abundant. In moist areas (axillae, groin, navel, and soles), bacteria of the genera *Staphylococcus* and *Corynebacterium*, including *S. epidermidis* and *S. aureus*, were detected.

In addition, across the entire skin body, the four major phyla of commensal bacteria were *Actinobacteria*, *Proteobacteria*, *Firmicutes*, and *Bacteroidetes*, while at least 19 phyla were found to be part of the bacterial skin microbiome.

### 3.3. Classification of Skin-Associated Bacteria (Favorable, Hazardous, and Opportunistic Bacteria)

These commensal skin bacteria form a stable flora by using cutaneous lipids and amino acids on the skin as a source of nutrients for their growth, and by establishing a competitive and harmonious relationship among themselves. Microorganisms in the skin flora are classified into those that contribute to health (favorable bacteria), those that contribute to disease (hazardous bacteria), and those that can be either (opportunistic bacteria) depending on the situation.

The representative good bacteria are *S. epidermidis*. In recent years, the use of *S. epidermidis* in the development of basic cosmetics tailored to individual skin conditions has attracted attention [16,17]. Forming indigenous flora on the skin surface and in the nasal cavity, *S. epidermidis* is known to produce glycerol and fatty acids by feeding on sebum and sweat (alkalinity) [17,18]. Glycerol moisturizes the skin and enhances the barrier function, while fatty acids keep the skin slightly acidic and prevent the growth of *S. aureus* by producing antimicrobial peptides [17,18]. Thus, *S. epidermidis* provides moisture to the skin, enhances the barrier function, and protects the skin from the proliferation of *Staphylococcus aureus.*

The typical hazardous bacteria are *S. aureus*. Present on skin surfaces and in pores, *S. aureus* is carried by about 30–50% of normal healthy adults [19].

Although normally harmless, it is highly pathogenic among staphylococci and is an important human bacterial pathogen responsible for a variety of conditions, ranging from

asymptomatic inflammation to severe infections causing pneumonia, endocarditis, and sepsis [19]. Preferring an alkaline environment, it proliferates when there is a low level of beneficial bacteria that maintain the skin's mild acidity, causing itching, rough skin, and atopic dermatitis. In addition to their tendency to settle on the skin of atopic dermatitis patients, the enterotoxins (toxins in the bacteria) produced by the bacteria exacerbate skin inflammation, reduce the skin barrier function, and are said to be a contributing factor to the worsening of symptoms [20,21]. In addition, if the injured skin is left untreated, it can fester and worsen. Thus, *S. aureus*, although harmful to skin health when overabundant, is necessary to maintain a balance of commensal bacteria.

A typical opportunistic bacterium is *C. acnes*. Previously, it was called *Propionibacterium acnes* because of its propionic acid-producing ability. *C. acnes* is an anaerobic bacterium that can hardly proliferate in an oxygenated environment. It exists in pores and sebaceous glands where there is no oxygen, and because it prefers lipids, it is especially abundant in the fat layer at the back of hair follicles (pores) in the skin, where sebum secretion is high [22,23]. It feeds on sebum to produce propionic acid, fatty acids [24], and keep the skin surface slightly acidic, thereby inhibiting the growth of pathogenic bacteria that adhere to the skin. Acne bacilli are generally considered to be the cause of acne, but if they do not proliferate excessively, they are good for the skin. However, when pores become blocked due to increased sebum production, epidermal keratinization, or inflammation, acne rods proliferate excessively, causing inflammation and leading to acne. In healthy skin, acne rods play a beneficial role in the skin microflora of the sebaceous gland unit of the hair follicle, but when they proliferate excessively, they become the causative agent of acne.

From the above, it is thought that indigenous skin bacteria do not normally exhibit pathogenic properties, but rather exert a kind of barrier function that prevents the entry of pathogens from the outside. However, when the barrier is broken or the balance between symbiotic organisms and pathogens is disrupted, skin and systemic diseases may occur.

In recent years, products have been developed in the cosmetics field that assist in the treatment of skin diseases by selectively sterilizing, eliminating, or inhibiting the growth of specific bacteria among these indigenous skin bacteria.

### 3.4. Health Functionality of Metabolites Produced by Indigenous Bacteria

It has been shown that indigenous skin bacteria grow on the components present on the skin as a source of nutrients. Research suggests that their breakdown products and metabolites have a health-related functionality, although epidemiological evidence is still to be established. For example, *C. acnes* secretes short-chain fatty acids (SCFA) and glycerol as products of abundant triacylglycerol fermentation in sebum. The major SCFAs are acetic acid, propionic acid, and butyric acid. Although the role of propionic acid in the skin is not yet understood, it maintains the pH of follicular sebaceous-gland hair follicles at an acidic level and limits the growth of *S. aureus* (such as community-acquired methicillin-resistant *S. aureus*) [25].

Glycerol has strong hygroscopic properties, which allow it to retain moisture down to the skin's stratum corneum and strengthen the skin's barrier function [26]. Therefore, *S. epidermidis* and *C. acnes* utilize endogenous carbon sources such as glycerol to produce SCFAs, such as acetic acid, butyric acid, and lactic succinate [27–29]. SCFAs have been reported to exhibit antibacterial activity [30] and inhibit the growth of *C. acnes* [27]. In particular, succinic acid effectively inhibited the growth of *S. acnes* in vitro and in vivo [27].

Previous studies demonstrated that butyrate from glycerol fermentation of *S. epidermidis* ATCC 12228 can downregulate ultraviolet (UV)-induced IL-6 secretion via activation of short-chain fatty acid receptor 2 (FFAR2) [31]. Butyrate has also been shown to function as an inhibitor of histone deacetylase (HDAC), which confers anti-inflammatory activity [32].

Lactate produced by epidermal bacteria inactivates HDAC11 on monocytes in the dermis of the skin, thereby activating HDAC6 and the production of IL-10, which suppresses immunity [33]. Lactic acid is also one of the natural moisturizing factors (NMF) of

the stratum corneum skin barrier, which helps to hydrate the skin surface and maintain a slightly acidic pH [34].

Free fatty acids in sebum are produced by the hydrolysis of triglycerides by lipases of indigenous skin bacteria. Sebum contains a variety of free fatty acids; oleic acid, linoleic acid, and linolenic acid are said to have antifungal effects on *Trichophyton rubrum*, while lauric acid and oleic acid inhibit the growth of *S. epidermidis* [35].

## 4. Application of Skin Probiotics in Cosmetics

Probiotics are defined as "live microorganisms that, when administered in appropriate amounts, provide health benefits to the host". Since they were proposed by Gibson in 1995, probiotics have been studied in earnest and are still widely used today for the treatment and prevention of gastrointestinal disorders. In recent years, not only the intestinal microflora but also the skin microflora have been studied from the viewpoint of probiotics, and their application to cosmetics has also been observed. In this chapter, we summarize the findings of skin probiotics that have been applied or are expected to be applied to cosmetics.

*Nitrosomonas eutropha* is an ammonia-oxidizing bacterium that produces NO and $NO_2$ on the skin through oxidation of ammonia contained in sweat [36]. In clinical trials, topical application of *N. eutropha* has been reported to improve wrinkles on the forehead and between the eyebrows, and to improve pigmentation [37]. Based on these results, a mist lotion containing *N. eutropha* is currently being commercialized, and the effects of this cosmetic have been confirmed as including improved skin clarity and skin texture.

Next, *Enterococcus faecalis* is a type of lactic acid bacteria that exists in the human intestine and is known to be beneficial to the body's health. Among them, *E. faecalis* SL5 has been reported as producing CBT-SL5, a kind of antibacterial peptide, and showing antibacterial activity against *C. acnes*. It has been confirmed as a possible new anti-inflammatory drug for acne [38]. In response to this, skincare cosmetics and other products utilizing *E. faecalis* have been commercialized and are formulated to suppress inflammation of acne.

*Vitreoscilla filiformis* is a fungus found in hydrothermal vents and hot spring water. Extracts of this fungus have been shown to balance the immune homeostasis of the skin. A study showed that ointments either containing or not containing *V. filiformis*, applied twice daily to the site of inflammation in patients with mild to moderate atopic dermatitis (AD) for 4 weeks, decreased the severity index (mEASI) and reduced the severity of pruritus. One study also confirmed a reduction in the severity of pruritus [39], while another showed that an extract of *V. filiformis* cultured in Vasi Volcanic Mineral Water (VVMW), known to strengthen the skin against exposome attack, was highly effective in restoring the skin barrier, preventing photoaging through enhanced immune defense, and repairing stressed skin. The results showed that *V. filiformis* is very effective in preventing photoaging and repairing stressed skin through restoring the skin barrier and enhancing immune defense [40]. In response, body washes and creams containing *V. filiformis* cultures have been commercialized and proven in clinical trials as effective in restoring the balance to the skin flora, as well as reducing itching in skin prone to allergies and atopic dermatitis [41].

*Streptococcus thermophilus*, a lactic acid bacteria known as the seed of yogurt and cheese, has been found to be beneficial in the skin. Studies have shown that, when creams containing *S. thermophilus* were applied to healthy subjects, ceramide levels, which are necessary for the barrier and moisturizing functions of the healthy stratum corneum, were increased [42]. Topical application to patients with atopic dermatitis (AD) also showed an increase in ceramide levels in the skin, and improvement of AD symptoms (erythema, scaling, and pruritus) was also confirmed [43]. Based on the above, *S. thermophilus* is used in cosmetics for its moisturizing and anti-inflammatory effects.

*Lactobacillus paracasei* is a lactic acid bacterium that exists in the intestinal tract and oral cavity, and is known to be beneficial to the skin, improving atopic dermatitis through oral intake [44]. In addition, when the effect of an extract of *Pueraria lobata* fermented with *L. paracasei* JS1 on the immune response of the skin was investigated, the expression of seven skin-related proteins was demonstrated, as well as a moisturizing effect on the skin. In

response to this, *L. paracasei* has been formulated for its moisturizing and anti-inflammatory effects. *L. paracasei* has also been reported to produce equol, which improves wrinkles, and may be used as a more beneficial skin probiotic [45].

*Lactobacillus rhamnosus*, which is expected to be applied to cosmetics as a skin probiotic, is also known to enhance the intestinal barrier function as a probiotic [46]. Furthermore, recent studies have revealed that topical application of *L. rhamnosus* lysate to a human epidermal model for 16 days promoted stratum corneum formation and increased the expression of skin barrier proteins, which may have beneficial effects on the maintenance of skin barrier function [47]. These results indicate the potential for use in skincare products as a topical probiotic, although additional studies are needed. In addition, *L. rhamnosus* showed a high inhibition of tyrosinase and melanin synthesis when co-cultured with the aforementioned *L. paracasei*, and increased collagen production compared to when cultured individually, indicating that co-culture may improve the efficacy of cosmetic products, such as skin whitening effects [48].

Functional foods that work similarly to probiotics include biogenics. Biogenics differ significantly from probiotics in that they act directly on the body without involving the intestinal flora. We also recently investigated the effects of lactic acid fermentation products (LFPs) on the skin. Oral administration of LFPs improved water transpiration and increased ceramide AP in a mouse model of atopic dermatitis [49]. These findings suggest that LFPs improve skin injury in an atopic dermatitis-like murine model. Moreover, LFPs were applied to three-dimensional cultured epidermis from the stratum corneum side. As a result, epidermal cell differentiation was promoted, the ceramide content and amino acid content were increased, and the amount of bound water in the stratum corneum was also increased [50]. These results suggest that LFPs play a significant role in moisturizing. These results indicate that oral or transdermal application of probiotics or biogenics may have a positive impact on maintaining skin health.

As described above, topical application of skin probiotics has been shown to not only improve acne and atopic dermatitis, but also have cosmetic effects in healthy skin, such as moisturizing and reducing skin aging.

## 5. The Japanese Cosmetics Industry and the Growth of Natural Ingredients and Fermented Cosmetics

In Japan, mercury, iron oxide, lead, and safflower have been used in cosmetics since the Heian period (A.D. 794–1185) and continued to exist until the Edo and early Meiji periods ([3], pp. 537–559). During the Edo period, Japan was closed to the outside world, trading only with the Netherlands, and the influx of Western culture was limited. However, with the Meiji Restoration of 1868 and the opening of Japan to Western culture, the custom of dying one's teeth black was abolished, and cosmetics using lead and mercury were discontinued as their harmful effects became well known through the spread of newspapers. The Meiji Restoration led to a capitalist economy, and from the mid-Meiji to Taisho periods, several cosmetics companies were established, with cosmetics using chemical products beginning to spread. The start of World War II led to restrictions on cosmetics as luxury goods, forcing the cosmetics industry to change its business model. The Pacific War caused the loss of Japan's industrial base, and after the war, some cosmetics companies disappeared, while others adopted American culture and started over. Some cosmetics companies also entered the market from other industries, and by the 1960s, the framework of the current cosmetics industry had been established.

Before the industrial evolution, cosmetics were made from plant-derived materials or minerals, such as lead, mercury, and iron oxide. However, with the establishment of the chemical industry in the early 20th century and the success of the pharmaceutical and plastic industries, most of the cosmetics industry also applied chemical products. Therefore, the use of chemical products in cosmetics has become a major trend. However, the disadvantages of using chemical products in cosmetics are now generally understood, and since the 1980s, cosmetics derived from plants and naturally derived materials such

as royal jelly have been growing. In fact, the market for natural skincare products is growing globally, expanding from USD 34.5 billion in 2018 to USD 36.3 billion in 2019 and is expected to reach USD 54.5 billion by 2027. In Japan, fermented microorganisms have been used in food production for more than 1300 years [51–54], and it has been elucidated that the ingredients themselves increase, or their activity is enhanced, through the fermentation process. In addition, there is a history of using fermented microorganisms in food production for over 1300 years in Japan. Many attempts have been made to apply them to cosmetics based on the understanding that the fermentation process increases the components of the ingredients that function as themselves, or enhances the activities of other components (such as kojic acid inducing the inhibition of tyrosinase activity [55], glucosylceramide increasing the expressions of tight junctions and ceramide delivery genes [56], enzymes involved in deglycosylation of isoflavone [57], and ethyl-$\alpha$-D-glucoside increasing collagen [58]).

## 6. Conclusions

This review introduced the application of fermented stuff-derived cosmetics and research involving skin microbiota. This new trend will innovate the cosmetics industry and improve skin-related welfare.

**Author Contributions:** Conceptualization, H.K.; validation, H.K. and A.O.; investigation, A.O., C.M., Y.S., K.T., N.H., S.I. and Y.T.; writing—original draft preparation, H.K., A.O. and Y.T., writing—review and editing, H.K. and A.O.; visualization, H.K.; supervision, H.K.; project administration, H.K.; funding acquisition, H.K. All authors have read and agreed to the published version of the manuscript.

**Funding:** This research received no external funding.

**Institutional Review Board Statement:** Not applicable.

**Informed Consent Statement:** Not applicable.

**Data Availability Statement:** Not applicable.

**Conflicts of Interest:** The authors declare no conflict of interest.

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
