# Peer review of "Fermented Cosmetics and Metabolites of Skin Microbiota—A New Approach to Skin Health"

_fermentation, doi:10.3390/fermentation8120703_

Round 1

Reviewer 1 Report

In this study, the manuscript summarized approaches to skin health using fermented cosmetics and modulation of the metabolites of skin microflora. Overall, the manuscript is technically sound and the review ideas appear justified.

However, the manuscript should be written introduction part. Moreover, “What are the advantages of Fermented cosmetics and metabolites in comparison with another approach to skin health”?

Listed are some comments regarding the submitted manuscript:

1.      Line 14: "largest organ of the human body →  "largest organ of the human body”.

2.      Staphylococcus epidermidis → S. epidermidis on all the text.

Author Response

Dear Reviewer1,

We really appreciate your logical review of our manuscript.

>However, the manuscript should be written introduction part.

References were added to the introduction part.

>Moreover, “What are the advantages of Fermented cosmetics and metabolites in comparison with another approach to skin health”?

This was added to text (L186). Epidemiological evidence is still not established.

>Listed are some comments regarding the submitted manuscript:

  1. Line 14: "largest organ of the human body →  "largest organ of the human body”. This was changed.
  2. Staphylococcus epidermidis → S. epidermidis on all the text. This was changed.

Reviewer 2 Report

The authors actually presented interesting paper and well-written manuscript but still there are few issues to be addressed.

1- from line 39 to line 64: although it is a very nice introduction on the skin structure and well-written but you did not mention any reference to these paragraphs.

2- line 107: you have repeated this sentence twice "The skin is in direct contact with the outside world" just remove one of them.

3- line 184: topic 3.4. you should explain this part more in details as it is very important to your paper work.

4- Topics: 3.5, 3.6, 3.7 should be subtitles from topic 3.4., please revise the order. 

Author Response

Dear Reviewer2, We really appreciate your logical review of our manuscript.

The authors actually presented interesting paper and well-written manuscript but still there are few issues to be addressed.

>1- from line 39 to line 64: although it is a very nice introduction on the skin structure and well-written but you did not mention any reference to these paragraphs. References were added.

>2- line 107: you have repeated this sentence twice "The skin is in direct contact with the outside world" just remove one of them. This was done.

3- line 184: topic 3.4. you should explain this part more in details as it is very important to your paper work. Epidemiological evidence is still not established in this area. This was added to L184.

4- Topics: 3.5, 3.6, 3.7 should be subtitles from topic 3.4., please revise the order.  This was done.